# Psychosocial singing interventions for the mental health and well-being of family carers of patients with cancer: results from a longitudinal controlled study

Daisy Fancourt,[1] Katey Warran,[2] Saoirse Finn,[2] Theresa Wiseman[3,4]

[1]Department of Behavioural Science and Health, University College London Research Department of Epidemiology and Public Health, London, UK
[2]Centre for Performance Science, Royal College of Music, London, UK
[3]Applied Health Research, The Royal Marsden NHS Foundation Trust, London, UK
[4]Faculty of Health Sciences, University of Southampton, Southampton, UK

**Correspondence to**
Dr Daisy Fancourt;
d.fancourt@ucl.ac.uk

## ABSTRACT

**Objective** The mental health challenges facing people who care for somebody with cancer are well documented. While many support interventions focus on provision of information or cognitive behavioural therapy, the literature suggests that psychosocial interventions could also be of value, especially given the low social support frequently reported by carers. Singing is a psychosocial activity shown to improve social support, increase positive emotions, and reduce fatigue and stress. This study explored whether weekly group singing can reduce anxiety, depression and well-being in cancer carers over a 6-month period.

**Design** A multisite non-randomised longitudinal controlled study.

**Setting** The Royal Marsden National Health Service Trust in Greater London.

**Participants** 62 adults who currently care for a spouse, relative or close friend with cancer who had not recently started any psychological therapy or medication.

**Interventions** On enrolment, participants selected to join a weekly community choir for 12 weeks (n=33) or continue with life as usual (n=29).

**Outcome measures** The primary outcome was mental health using the Hospital Anxiety and Depression Scale. The secondary outcome was well-being using the Warwick Edinburgh Mental Wellbeing Scale. Using linear mixed effects models, we compared the change in mental health and well-being over time between the two groups while adjusting for confounding variables including demographics, health-related variables, musical engagement and length of time caring.

**Results** Participants in the choir group showed a significantly greater decrease in anxiety over time than participants in the control group (B=−0.94, SE=0.38, p=0.013) and a significantly greater increase in well-being (B=1.25, SE=0.49, p=0.011). No changes were found for depression. Sub-group analyses showed carers with anxiety or below-average well-being were most likely to benefit.

**Conclusions** This study builds on previous research showing the mental health benefits of singing for people with cancer by showing that weekly singing can also support anxiety and well-being in cancer carers.

## Strengths and limitations of this study

► This study had many strengths including its longitudinal tracking involving multiple time-points and its controlled design.

► We used random-intercept random-slope linear mixed effects models which allowed us to compare different rates of change in mental health and well-being over time among carers who did and did not take part in a weekly choir over 24 weeks.

► Our results were found independent of identified confounding factors including demographics, health-related variables, other musical engagement and length of time caring.

► Our groups were statistically well matched at baseline, and results were not attenuated by the consideration of baseline confounders.

► However, our results were found in a relatively modest sample size, so future studies are required to ascertain whether results can be replicated in larger samples, and our study was not randomised so we cannot assume full exchangeability between groups.

## INTRODUCTION

Family members and close friends, particularly spouses, are often primary caregivers for individuals affected by cancer, playing a vital role in their care. Yet, well documented across the literature are the psychological, physical, social and economic challenges faced by informal cancer carers.[1 2] In relation to mental health, carers have repeatedly been found to have greater anxiety and depression levels than general population controls.[3 4] Indeed, psychological distress in carers correlates with the distress experienced by patients and has been found through meta-analysis to be no different in size.[5 6] Prevalence rates of depression are around 26% in both patients and spouses, and the prevalence of anxiety is around 28% in patients and 40% in spouses.[7] In particular,

anxiety has been found to increase when caring for somebody with advanced cancer, exceeding depression in carers.[8] This has led to suggestions that anxiety, rather than depression, is the largest psychological problem in carers and calls for interventions that can tackle it.[7]

In addition to mental health challenges, caring for somebody with cancer has also been found to affect well-being. A Europe-wide survey found mental as well as physical health-related quality of life to be significantly lower in carers than non-caregivers.[9] Female carers in particular report lower quality of life than women who are not carers,[5] and quality of life for carers of both genders has been found to vary across the illness trajectory.[10] Given that well-being is associated with mental health and also related factors such as resilience and coping, there is a need too for interventions that can support well-being in carers.

Many interventions for carers focus on the provision of information designed to directly support carers in their roles or involve cognitive behavioural therapy (CBT) to support and enhance coping.[11 12] A meta-analysis of therapeutic counselling, skills training and psychoeducational interventions for caregivers of cancer patients found evidence of reductions in caregiving burden, enhanced coping strategies, self-efficacy, physical functioning, anxiety, family relations and social functioning.[13] Interventions were identified as being more effective if caregivers attended without the person they cared for and if the interventions were delivered in a group setting with other carers. In addition, there has also been increasing interest in the role of psychosocial interventions for supporting mental health and well-being in carers.[14] Such interest has been encouraged by research suggesting that, in addition to specific needs relating to their caregiving, carers frequently have unmet personal needs and low social support themselves.[15] In a population-based study of family carers of cancer survivors, social support was one of the strongest predictors of quality of life.[16] Carers with less social support also have a higher risk of developing depression and anxiety.[1] Yet, caregivers often feel that they receive less support in comparison to how much patients feel that they receive.[16] So there is a need for research into group-based psychosocial interventions that could improve mental health and wellbeing in carers.

Singing is a promising multimodal psychosocial intervention that has been used to support mental health and well-being in diverse populations, including among patients with cancer.[17] In cancer caregivers, preliminary research has suggested that singing can increase perceived social support, enhance short-term positive emotions, reduce fatigue, promote relaxation and also reduce levels of stress hormones and mood-related inflammation over a single 1-hour singing session.[18] However, to date, there is little research into whether singing can specifically reduce anxiety and depression and enhance well-being in cancer caregivers over longer periods of time. While a few studies have identified benefits of singing for other groups of carers, such as carers of people with dementia, these studies have typically been very small in sample size (e.g., four to seven participants completing data collection), followed participants over relatively short time frames (e.g., just six weeks) and employed little quantitative data.[19–22] Consequently, this study built on the theoretical rationale for the benefits of psychosocial interventions for cancer carers and the preliminary data showing the benefits of singing specifically for carer populations and explored whether weekly singing is linked with improvements in anxiety, depression and well-being in carers of somebody affected by cancer over a six month period. We hypothesised that singing in a choir would be associated with improvements in all three outcomes, and that this response would be strongest among those with the poorest mental health and well-being at baseline.

## MATERIALS AND METHODS
### Participants and procedure

This study was a multisite non-randomised longitudinal controlled study. To be eligible, participants had to self-identify as currently caring for a spouse, relative or close friend with cancer. We excluded members of hospital staff to differentiate between formal and informal care. Participants were excluded if they were under the age of 18, if they were already engaged in a weekly group choir, if they had started a formal course of psychological therapy in the last month or were scheduled to start one in the next 12 weeks, if they had started any new medication for anxiety or depression in the last month or if their level of English was insufficient to complete the questionnaires required. Participants were recruited through National Health Service (NHS) hospital trusts in Greater London, carer support groups, community and charity events, and social media. A total of 396 participants were screened for enrolment, 66 were eligible and consented to take part, and 62 participants (29 control and 33 experimental) enrolled in the study (see figure 1 for a CONSORT diagram of participants).

On enrolment, participants were given the option of joining a weekly choir for 12 weeks. This length of study was selected from pilot data collected on singing for patients affected by cancer who were involved in uncontrolled studies in Wales, which suggested that improvements in mental health were found 12 weeks following joining, with further improvements by 24 weeks.[23] Participants who did not select to join a choir formed the control group and continued with life as usual. Participants who did select to join a choir took part in a 90 minute session each week led by a professional choir leader, of which 60 minutes was dedicated to singing (including warm-up exercises, learning new songs and singing familiar songs) and 30 minutes to socialising. The repertoire focused on popular songs arranged specifically for the choirs with backing tracks. All songs were learnt by ear without participants needing to read music. These choirs were led by Tenovus Cancer Care, specifically for people affected in some way by cancer. The format of the sessions has been

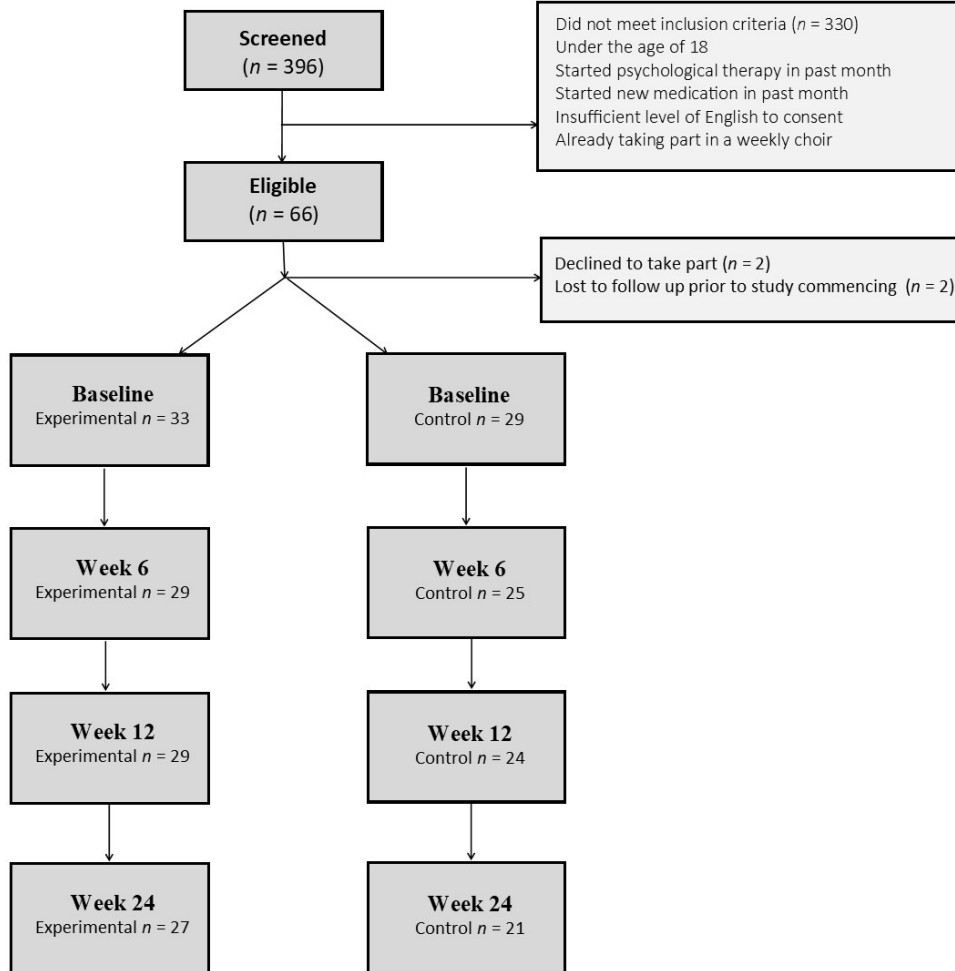

**Figure 1** CONSORT diagram of participants.

described in previous studies.[24] Both choirs involved in this study were established specifically for the purposes of collecting data for this research. Following 12 weeks in the choir, participants were given the option to continue with weekly choir sessions if they wished, with 19 participants (58%) choosing to leave and the remaining 42% choosing to continue attending. Those who continued attending attended an average of 10 sessions out of a further 12, with 57% of them attending all 12. All participants were asked to complete questionnaires at baseline, 6 and 12 weeks as well as at 3-month follow-up (24 weeks)

### Patient and public involvement

The research questions and outcome measures were selected with the support of a Patient and Public Involvement (PPI) group of participants in the Tenovus Cancer Choirs in Wales. These participants also approved the design of the study. As this was the first study of the choirs at the Royal Marsden Hospital sites (or indeed in England) and there was therefore not data about the response from individuals in the community or the hospitals to the choirs, it was proposed by our PPI group that the study could be run as a longitudinal controlled study rather than a randomised controlled trial (RCT). As such, any carers would have the option of joining immediately

rather than becoming part of a wait-list control. While this has obvious methodological limitations, it was felt to be an important precursor to an RCT that would enable an assessment of the reception of the choirs in these new locations alongside measurements of outcome data. Participants in the study also supported recruitment efforts by helping to identify further participants. Dissemination plans include presentations back to members of all of the Tenovus Cancer Choirs in Wales and England.

### Measures

Our primary outcomes were anxiety and depression, measured using the Hospital Anxiety and Depression Scale (HADS), which contains two independent subscales: one for anxiety and one for depression ranging from 0 to 21 for each construct with higher scores indicating poorer mental health (Bjelland et al, 2002).[25] When using this measure, scores of 0–7 are considered as non-indicative of anxiety or depression, while scores of 8–10 present mild cases, 11–14 present moderate cases and 15–21 present severe cases.[26]

Our secondary outcome was well-being, measured using the Warwick Edinburgh Mental Wellbeing Scale (WEMWBS) short form, which encompasses both hedonic and eudemonic well-being.[27] The scale is scored

from 7 to 35 with higher scores representing higher levels of well-being. As recommended in validations, the raw scores were logit transformed prior to analysis.[28] The New Economics Foundation suggests five levels of well-being based on quintile analyses of data in the UK Understanding Society Survey, 2009: poor (<22), below average (22–24), average (25–26), good (27–28) and excellent (29–35).

Additionally, we collected data on age, gender, income (<£16,000, £16,000-£30,000, £31,000-£60,000, £61,000-£90,000, >£90,000), employment status (unemployed, voluntary/temporary work, part-time work, full-time work, retired), whether participants were receiving psychological therapy, whether participants had any other health condition, whether participants had previously sung in a choir, how confident participants felt about singing (from 1-not at all to 5-very), whether participants had attended musical concerts or performances in the past year, whether they had taken part in any other music activity in the past year, and how long participants had been in a caring role for.

## Statistics

To compare the choir versus the control group at baseline, we used one-way analyses of variance and Fisher's exact tests. To explore changes in mental health and well-being over time and between groups, we used linear mixed effects models (LMMs). LMMs allow for natural heterogeneity among participants and, unlike some other repeated measures analysis models such as repeated measures analyses of variance which deal with missing data through list-wise deletion, linear mixed effects models make full use of the dataset. Our participants provided data an average of 3.4 times across the 4 time points, providing 180 data points. Our models used a random intercept and random slope, with an unstructured covariance matrix of the random effects. Time was modelled first as both a continuous variable to identify an overall time*group relationship (using a linear model as we found no evidence to support the use of a quadratic model), and then time was modelled as four separate time points to identify where specific changes occurred. Standardised residuals were generated to confirm the assumption of normality and there was no evidence of heteroscedasticity.

We built up our final model through considering nested models of relevant covariates. Model 1 was unadjusted apart from baseline mental health, model 2 adjusted for demographic variables (age, gender, income and employment status), model 3 additionally adjusted for health-related variables (undergoing psychological therapy and any other health conditions), model 4 additionally adjusted for cultural engagement and attitudes to singing (past experiences singing in a choir, confidence in singing, attendance at concerts or performances in the past year and engagement in any other musical activity in the past year) and model 5 additionally adjusted for length of time caring for somebody with cancer. We used

the log-likelihood ratio, Akaike's information criterion and Bayesian information criterion to confirm best fit of model 5. We calculated margins of response from the fully adjusted model and created profile plots to illustrate the time*group interactions. As three outcomes were considered, a Bonferroni's alpha of 0.05/3=0.017 can be assumed.

To ascertain whether those with poorer mental health and well-being were more or less likely to benefit from engagement, subgroup analyses excluded those with no evident mental health problems at baseline. Additionally, given participants had been given the option of whether to continue with weekly singing following the first 12 weeks, we statistically compared differences in demographics and mental health among those who did and did not continue and also ran sensitivity analyses additionally adjusting for singing patterns after the initial 12 weeks to see if these affected outcomes at 24 weeks. All results shown are fully adjusted models (i.e., model 5), and all analyses were conducted using Stata V.14.

## RESULTS

### Demographics

Participants were well matched at baseline on all demographic variables including mental health and well-being, with no difference in baseline levels of anxiety, depression or well-being. The only significant difference between groups was in confidence in singing, with the control group showing lower overall confidence than the singing group (see table 1).

### Anxiety

At baseline, 44.3% (n=27) of participants showed no anxiety (HADS Anxiety (HADSA) score 0–7), 21.3% (n=13) showed mild anxiety (HADSA score 8–10), 23.0% (n=14) showed moderate anxiety (HADSA score 11–14) and 11.5% (n=7) showed severe anxiety (HADSA score 15–21). One participant in the control group was missing data on baseline anxiety. The linear mixed effects model with time as a continuous variable showed that there was a significant time by group interaction, with participants in the choir group showing a significantly greater decrease in anxiety than participants in the control group (B=−0.94, SE=0.38, p=0.013). When modelling time as four separate time points, this difference was not apparent in the first 6 weeks (B=0.22, SE=0.81, p=0.78) but was apparent by week 12 (B=−3.06, SE=0.92, p=0.001), with the difference between groups decreasing slightly by week 24 (B=−1.93, SE=1.12, p=0.086) (see figure 2A). In the choir group, this equated to 17.5% decrease in anxiety symptoms across the first 12 weeks, which had slightly decreased to an 11.9% decrease in anxiety symptoms by 24 weeks.

### Depression

At baseline, just 14.8% (n=9) of participants showed mild depression and only 11.5% (n=7) showed moderate depression, with no participants showing severe

**Table 1** Baseline demographics of participants

| | Control (n=29) | Choir (n=33) | P value |
|---|---|---|---|
| Age, mean (SD) | 51 (15) | 58 (14) | 0.075 * |
| Sex, % female | 72.4 | 81.8 | 0.54† |
| Income, % | | | 0.076† |
| <£16 000 | 12.0 | 20.7 | |
| £16,000-£30 000 | 24.0 | 41.4 | |
| £31,000-£60 000 | 32.0 | 24.1 | |
| £61,000-£90 000 | 32.0 | 6.9 | |
| >£91 000 | 0 | 6.9 | |
| Employment status, % | | | 0.22† |
| Unemployed | 13.8 | 3.2 | |
| Voluntary | 0 | 6.5 | |
| Part-time work | 20.7 | 29.0 | |
| Full-time work | 37.9 | 22.6 | |
| Retired | 27.6 | 38.7 | |
| Currently having therapy, % | 10.3 | 24.2 | 0.19† |
| Health condition, % | 72.4 | 60.6 | 0.42† |
| Previously sung in a choir, % | 65.5 | 48.5 | 0.21† |
| Confidence in singing, % | | | **0.040**† |
| 1—not at all | 50 | 21.2 | |
| 2 | 21.4 | 12.1 | |
| 3 | 21.4 | 36.4 | |
| 4 | 3.6 | 21.2 | |
| 5—very | 3.6 | 9.1 | |
| Attended a concert or performance in the past month, % | 32.1 | 51.5 | 0.19† |
| Took part in a music activity in the past month, % | 18.5 | 36.4 | 0.16† |
| Length of time being a carer, % | | | 0.45† |
| 0–3 months | 0 | 9.7 | |
| 3–12 months | 17.9 | 19.4 | |
| 1–3 years | 50 | 35.5 | |
| 3–5 years | 14.3 | 22.6 | |
| 5+ years | 17.9 | 12.9 | |
| Anxiety, mean (SD) | 8.04 (4.71) | 8.97 (4.19) | 0.42* |
| Depression, mean (SD) | 5.43 (4.49) | 4.88 (3.08) | 0.57* |
| Well-being, mean (SD) | 23.15 (4.36) | 21.71 (3.70) | 0.17* |

*One-way ANOVA.
†Fisher's exact test.

depression. There was no evidence of a time by group interaction for depression with time as a continuous variable (B=−0.49, SE=0.39, p=0.21). When modelling time as four separate time points, there was similarly no difference evident at 6 weeks (B=−0.18, SE=0.81, p=0.82), 12 weeks (B=−1.17, SE=0.94, p=0.22) or 24 weeks (B=−1.28, SE=1.19, p=0.28) (see figure 2B).

## Well-being

At baseline, 52.5% (n=32) of participants had poor well-being (<22), 16.4% (n=10) were below average (22–24),

16.4% (n=10) were average (25–26), 8.2% (n=5) were good (27–28) and 6.6% (n=4) were excellent (29–35). One participant in the control group was missing data on baseline wellbeing. There was a significant time by group interaction for well-being when modelling time as a continuous variable, with participants in the choir group showing a significantly greater increase in well-being than participants in the control group (B=1.25, SE=0.49, p=0.011). When modelling time as four separate time points, this difference was not apparent in the first

A

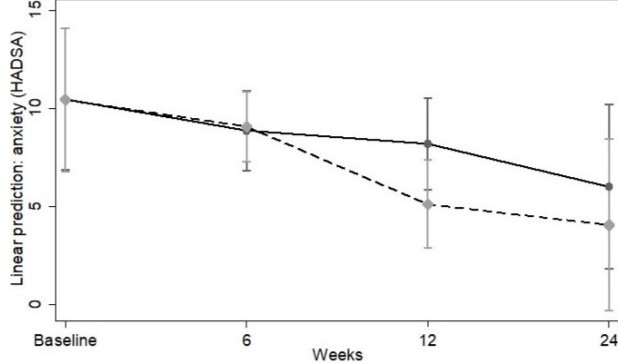

B

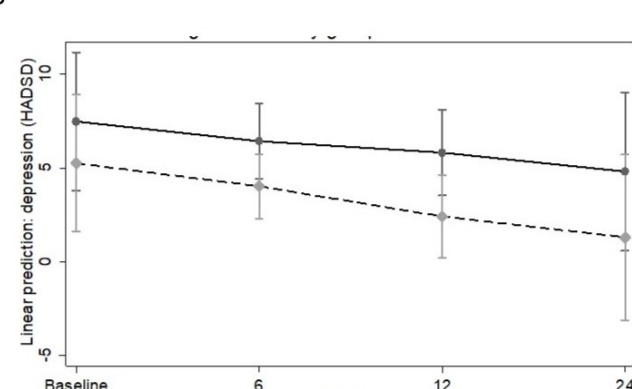

C

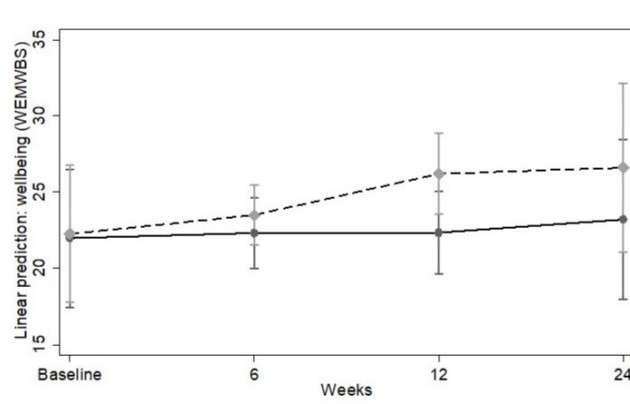

**Figure 2** Predictive margins of time by group interaction with 95% CIs for (A) symptoms of anxiety, (B) symptoms of depression, (C) well-being. Control group is shown as a hard line, and experimental group as a dashed line. The model is adjusted for demographic factors, health-related factors, cultural engagement and attitudes to singing and length of time caring for somebody with cancer. HADS, Hospital Anxiety (HADSA) and Depression Scale (HADSD); WEMWBS, Warwick Edinburgh Mental Wellbeing Scale.

6 weeks (B=0.89, SE=0.98, p=0.36) but was apparent by week 12 (B=3.58, SE=1.17, p=0.002) and was still present by week 24 (B=3.10, SE=1.50, p=0.038) (figure 2C). This equated to a 12.6% increase in well-being in the choir group across the first 12 weeks, which had slightly decreased to a 10.5% increase by week 24.

### Subgroup and sensitivity analyses

We used subgroup analyses to test whether participants in the choir group demonstrating anxiety and/or low well-being at baseline showed improvements over the 24 weeks compared with those in the control group.

### Anxiety

As a first subgroup analysis, we excluded participants with no evidence of anxiety at baseline (HADSA score <8) and reran analyses (n=34; control group=16, choir group=18). This confirmed a significant time by group interaction for anxiety, with participants in the choir group showing a significantly greater decrease in anxiety than participants in the control group (B=−2.62, SE=0.58, p<0.001). This difference was not apparent in the first 6 weeks, but was apparent by week 12 (B=−7.47, SE=1.28, p<0.001) and was maintained at week 24 (B=−5.57, SE=1.64, p=0.001).

These participants with anxiety also showed evidence of improvements in well-being, with a significant time by group interaction (B=1.93, SE=0.57, p=0.001). As with full-sample well-being analyses, this difference was not apparent in the first 6 weeks (B=0.25, SE=1.08, p=0.81) but was apparent by week 12 (B=5.12, SE=1.27, p<0.001) and was maintained at week 24 (B=4.32, SE=1.68, p=0.01).

### Well-being

As a second subgroup analysis, we excluded participants with average or above-average well-being at baseline (WEMWBS score >25) and reran analyses (n=42; control group=18, choir group=24). This confirmed a significant time by group interaction for well-being, with participants in the choir group showing a significantly greater improvement in well-being than participants in the control group (B=1.30, SE=0.51, p=0.01). This difference was not apparent in the first 6 weeks (B=1.45, SE=1.00, p=0.15) but was apparent by week 12 (B=1.45, SE=1.16, p<0.001), although it was slightly attenuated at week 24 (B=2.59, SE=1.52, p=0.087).

These participants with low well-being also showed evidence of improvements in anxiety, with a significant time by group interaction (B=−1.22, SE=0.51, p=0.017). As with full-sample anxiety analyses, this difference was not apparent in the first 6 weeks (B=0.95, SE=0.96, p=0.33) but was apparent by week 12 (B=−3.91, SE=1.10, p<0.001), although it was slightly attenuated by week 24 (B=−2.34, SE=1.41, p=0.097).

### Singing continuation

We compared differences between those who did and did not continue with singing beyond the first 12 weeks. There were no demographic differences between groups or differences in mental health between those who continued with the singing following week 12. As a sensitivity analysis, we additionally factored in whether participants continued to sing beyond the first 12 weeks on our primary and secondary outcomes. Singing behaviours after the initial 12 weeks were found to not affect the

significance of overall results nor significantly predict trajectories in anxiety, depression or well-being.

## DISCUSSION

This study explored whether singing in a choir is associated with improvements in mental health and well-being in people who care for somebody with cancer. Results partly confirmed our hypotheses, showing that singing in a choir on a weekly basis was significantly associated with improvements in anxiety and well-being, but not depression. These results were independent of demographic covariates, health-related covariates, previous or current engagement in musical activities, attitudes to singing, and length of time caring. As predicted, these improvements in mental health and well-being were found among carers generally, as well as specifically carers with anxiety or below-average well-being. Differences between groups were not found within the first 6 weeks but were found by week 12 and were generally maintained at the 3-month follow-up. Forty-two per cent of carers involved in the choir group continued to sing in the follow-up period. However, continuing to sing beyond the initial 12-week period did not significantly predict well-being or anxiety at follow-up.

Our findings are in line with meta-analyses suggesting that anxiety rather than depression is the major challenge among cancer carers.[7] In total, 56% of participants showed symptoms of anxiety, with 35% specifically showing symptoms of moderate or severe anxiety. The finding that singing was associated with a greater decrease in anxiety echoes findings from previous studies of singing among people affected by cancer. In a study of 143 non-patients affected by cancer (comprising carers, staff and those who had been bereaved), singing reduced levels of anxiety but not depression using HADS over a 6-month period.[23] Further, our finding suggesting benefits continued to be felt regardless of whether participants sung after the initial 12 weeks mirrors results from group drumming interventions for people with mild–moderate mental health conditions, which similarly found a maintenance of mental health benefits 3 months following the end of a 10-week programme.[29]

A key consideration is how choir singing might reduce anxiety. A separate grounded theory study of the choirs involved in this study identified four key mechanisms.[30] First, the choirs provided emotional and uplifting experiences that participants experienced as a mind–body activity that supported their sense of identity. Second, the choirs provided social support; specifically an 'unspoken' group support that enabled the development of friendships, the perception of a caring network and a sense of inspiration from the choir leader. Third, the choirs helped to build resilience among participants, developing coping skills, building confidence and leading to wider social and behavioural changes in the lives of individuals beyond the rehearsals themselves. Finally, the choirs provided members with the chance to develop

musical skills which enhanced their sense of self-esteem. In particular, the social support provided by the choirs may have been a key component for the carers in this study, as social support has been found to be a significant predictor of anxiety and quality of life in caregivers.[1 16]

### Strengths and limitations

This study had many strengths including its longitudinal tracking involving multiple time-points and its controlled design. We also considered factors such as additional psychological support being received by participants and deliberately only included participants who had not recently had any change in their psychological support or medication. The study was not randomised, so while this did have the advantage of mimicking real-life choice in that participants had the option of selecting whether to enrol in the choir or not, it means that causality cannot be assumed. Notably, we used a statistical approach that allowed us to model the effects of variables that could have affected exchangeability between groups on mental health trajectories across the 24 weeks. Our groups were extremely well matched statistically at baseline, and results were not attenuated by the consideration of baseline confounders. But future RCTs are required to confirm whether involvement in choirs is causally linked with mental health. Further, there was a significant difference at baseline in terms of confidence in singing with those who joined the choir professing to be more confident. While we factored this difference into our analyses such that it cannot explain the results, it does suggest that singing interventions may not be suitable for all carers or may need adaptations to attract and retain those who feel less confident. Our results were also found in a relatively modest sample size, so future studies are required to ascertain whether results can be replicated in larger samples. Future studies could benefit from focusing more specifically on particular points in caregiving (such as around initial diagnosis or during palliative care). Such studies could also focus specifically on carers with known mental health diagnoses such as moderate anxiety, especially given the findings here that singing was of benefit both to those with and without baseline anxiety or low well-being. Finally, this study focused exclusively on carers of somebody with cancer. However, whether the longitudinal benefits of group singing suggested here are replicated among other carer groups remains to be explored.

### CONCLUSIONS

In conclusion, this study builds on previous research showing the mental health benefits of singing for people affected by cancer by showing that weekly singing may also support anxiety and well-being in carers. Given that singing can be provided as a community activity free from the stigma that sometimes surrounds engagement with more formal support services, it could be promoted to support the mental health and well-being of cancer

caregivers. These results suggests the value of conducting future randomised controlled studies.

**Contributors** DF and TW designed the study. SF and KW collected the data. DF analysed the data and drafted the manuscript. All authors critically appraised the manuscript and approved it for publication.

**Funding** This research was funded by Tenovus Cancer Care [SWU2016-01]. DF is funded by the Wellcome Trust [205407/Z/16/Z]. TW is funded by the Royal Marsden Cancer Charity. The research was facilitated by the infrastructure of the RM/ICR NIHR Biomedical Research Centre. The study team acknowledge the support of the National Institute of Health Research Clinical Research Network (NIHR CRN).

**Competing interests** The funding for this study was provided by Tenovus Cancer Care who run the choirs used as the intervention in this study. However, no member of staff from Tenovus Cancer Care was involved in the specifics of the study design or in the data collection, analysis or interpretation of the data, the writing of the report, or the decision to submit the paper for publication.

**Patient consent for publication** Not required.

**Ethics approval** The study received ethical approval from the National Research Ethics Service [16/LO/0579] and all participants provided informed consent.

**Provenance and peer review** Not commissioned; externally peer reviewed.

**Data availability statement** Data are available upon reasonable request.

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
