## [Reviewer comments · BMJ Open]

ARTICLE DETAILS

TITLE (PROVISIONAL)	Psychosocial singing interventions for the mental health and wellbeing of family carers of cancer patients: results from a longitudinal controlled study
AUTHORS	Fancourt, Daisy; Warran, Katey; Finn, Saoirse; Wiseman, Theresa

VERSION 1 - REVIEW

REVIEWER	Jill Cameron University of Toronto, Canada
REVIEW RETURNED	31-Oct-2018

GENERAL COMMENTS	This is an unique study considering the benefits of singing to the mental health of carers to individuals with cancer. It is clearly written and the importance of studying anxiety in this population is well justified. Findings suggest carers anxiety and wellbeing improve as a result of the singing intervention and that those who were in worse mental health benefit more. Consideration of the following could further strengthen the message of the manuscript. In the introduction, 3rd paragraph, you suggest that group versus individual intervention moderated effects on the outcome - can you please explain the moderating effect? Randomization was not included in the study design due to the input of the Patient and Public Involvement group - can you explain their concern with using randomization and how this decision was reached? Part of the rationale for this study was the need for a longer-term follow up to examine long term benefits of the intervention. Please justify the choice of a 3-month follow up period and why this is considered long-term. With respect to the anxiety measure, what is the justification for the specified categorization as mild, moderate or severe? The statistical analyses are well described and appropriate to address the research objective. Can you provide a sample size estimate for the univariable, multivariable, and sub-group analyses? In table 1, considering the small sample size, please consider collapsing some of the categorical responses (e.g., income, employment status, singing confidence, length of time as carer, etc) and re-examine group equivalency. Also, clarify how these variables were included in the multivariable models. Were they included as is or in collapsed groups?
--

	Clarify if the presentation of the results are based on the univariable or multivariable analyses. It would be beneficial to present these results in table form. Please explain why the sensitivity analyses were conducted and why these specific analyses. With respect to the CONSORT diagram (Figure 1), please expand to provide more detail about the 330 potential participants who did not meet inclusion / exclusion criteria. What criteria did they not meet? Also include reasons participants were lost to follow up. Figure 2, please clarify if this is a presentation of the unadjusted or adjusted analyses. It is not clear how significant group differences were identified when there are overlapping confidence intervals.
--	---

REVIEWER	Dr. Zoe Hoare NORTH CTU Bangor University UK
REVIEW RETURNED	31-Jan-2019

GENERAL COMMENTS	The concept of using singing therapies is not new and has been evaluated and utilised in many participant groups. Unfortunately I think the design of this study has not contributed to the addition of new knowledge to the literature. There was no justification of why carers for cancer patients in particular rather than carers more generally, as this type of intervention has been used in dementia 'Singing for the Brain', 'Singing together'. Was this just a convenience sample? For me the glaring limitation of the design is in relation to not using randomisation, there may have been justified reasons for this but these are not clearly described in the paper alluding that PPI input agreed that the design should not be randomised is not enough. There are a number of designs that could have been considered that would have allowed consideration of participant preference. Allowing the participants to 'choose' their group will have significantly biased the estimate of effect as those who chose the intervention would have been motivated differently to those who did not (although it was interesting to see that the numbers who had had previous choral experience were higher in the control arm - possibly indicating that this type of intervention really wasn't for them). In my opinion, I don't think this bias is analogous with 'mimicking real life choice' as the effect estimate. In a pragmatic randomised design the effect estimate would be representative of the effectiveness of the intervention when offered to the specified population, however this preference design is self-selecting and potentially drastically increases the effect estimate produced. Those who choose not to partake in the intervention are more 'extreme' to the possible advantage of the intervention than those possibly who would have signed up for a randomised study and been assigned an arm not particularly of their preference, i.e. allowing the preference has included participants that would never have signed up for this type of evaluation if it were randomised. Comparison of baseline factors to establish comparability does not take into account this preference/motivation difference. The sample size used appears to be convenience rather than justified and although acknowledged in the limitations explanation
---

	of the use of the sample would be useful in understanding the context of the design. Given the design it is reassuring to see that the models used to elicit the results include the known confounders (although given the model building technique this was not by plan it seems) and I agree that the statistical analysis used is appropriate for the data. Whilst some statistical significance is found there appears to be no relation of this statistical finding to clinical impact? Given the small sample it is likely that this impact will be of clinical significance but in reporting this needs to be drawn out, particularly as this estimate is likely to be biased given the design. The sensitivity analysis given in the paper would be better framed as sub-group analysis than sensitivity analysis as this is essentially what it is and given the sample size is even more limited in the conclusions that can be drawn. Although dropout from the study is considered and missing data accounted for via the statistical model one thing that has not been described or accounted for is attendance /adherence to the intervention. The assumption currently is that all participants who provide data until the 24 week follow up attend all sessions of the provided intervention. I don't agree that a strength of the study should be highlighted as using validated measures as it is unlikely that anyother type of measure would be used. I appreciate that the design is as given now and very little can be done about this and there are a number of justified reasons as to why this design would be appropriate. I do accept that a rigorously conducted longitudinal study can provide good evidence (even more than a poorly conducted randomised design in some circumstances) but the justification and reasoning needs to be clear in the reporting. If the design choices relating to the design and sample size can be justified then this will greatly help the explanation of the study findings, with appropriate addressing of the limitations.
--	--

REVIEWER	Nick Garrett PhD Auckland University of Technology, Auckland, New Zealand
REVIEW RETURNED	08-Feb-2019

GENERAL COMMENTS	The article is well written (one typo one page 8 line 176 'different' should be 'difference') however the biggest design issue is the lack of randomisation between the intervention and control groups. This in itself while not ideal is acceptable and as the authors state mimicks real life, but needs to be more explicit stated in the abstract (just stated 'selected to join ...' is understating the lack of randomisation) and interpretation of results not just in the limitations at the end and especially as it is not stated in the abstract limitations. It is not clear in the abstract or in the measures section that the HADS has both anxiety and depression subscales, it is only by
--

	either prior knowledge or inference by the fact that anxiety and depression are reported separately. There is a large reduction from those screened to the number that were eligible. It would be useful for readers to know the impact of each of the eligibility criteria on numbers i.e. whether it was therapy, medication, English language, or already singing. A number of participants (42%) continued on with the choir, while it appears to have no impact based on the results, it would be ideal to know whether all of them continued past the 3 month follow-up point or not. If they did not all continue past the 3 month assessment then this would reduce the impact of no significant effect at 3 months. I also note that no numbers or statistics are reported for the difference for those that continued vs those that did not, with the reduction in numbers at this point the size of the effect is most probably more important than its significance. The statistics section of the methodology states that "We modelled time both as a continuous variable and as a four separate time points". Is this in the same model, or separate models, it is not clear. Also the results specify time effects and group x time interactions, which time variable are these? The table of the baseline demographics of participants shows a number of zero cells for the control group i.e. income >91,000, voluntary work and 0-3 months being a carer. These zero cells would have an impact on the appropriateness of a Chi-square test. Either a collapsing of categories or use of a Fisher's Exact test would be appropriate, and may have an impact on the significance of the test. It is interesting to note that the most wealthy, volunteers, and recent carers all selected to be in the intervention. These factors along with the confidence in signing, all show something slightly different about the intervention group. The use of confounders in the model building was confusing, the methodology refers to the building up of the final model with demographics, health related, cultural engagement and attitudes to singing, and length of time caring. However nowhere is there reference to exactly which factors were actually included in the final models. Is it all the baseline factors regardless of whether they had statistically significant results or only a select few. Or were other factors not in Table 1 utilised this is not clear. All of the linear mixed effects models are presented in text only, and it is difficult to make sense of the different components of the model presented in the text. There appears to be an overall time x group interaction effect. Then some time point by time point effects/statistics are examined, however this is not referred to in the methodology. Similar format is used in the sensitivity analysis. For the strengths and limitations in the article summary, the first two dot points are design not strengths or limitations of the study.
--	--

REVIEWER	Stelios Zimeras University of the Aegean Department of Statistics and Actuarial-Financial Mathematics
REVIEW RETURNED	14-Feb-2019

GENERAL COMMENTS	The paper is interesting. Some statistical suggestions must firstly considered  1. The authors have not include analysis of normality. They consider probably normal distributions for their data that it is not correct. The have applied parametric statistical analysis but they have not investigate if their data are ok for that. 2. The number of patient are very low. 29 for control and 33 for choir. They have not investigate the sampling significance of their data. They are ok for their analysis. 3. Based on their data the reader would wait an logistic regression analysis or OR analysis
--

VERSION 1 – AUTHOR RESPONSE

Response to reviewers

Reviewer: 1

Reviewer Name: Jill Cameron

Institution and Country: University of Toronto, Canada Please state any competing interests or state 'None declared': None declared

This is an unique study considering the benefits of singing to the mental health of carers to individuals with cancer. It is clearly written and the importance of studying anxiety in this population is well justified. Findings suggest carers anxiety and wellbeing improve as a result of the singing intervention and that those who were in worse mental health benefit more.

We are pleased that Dr Cameron found the study of importance and clearly written. We are grateful for the comments below, all of which have been addressed.

- Consideration of the following could further strengthen the message of the manuscript. In the introduction, 3rd paragraph, you suggest that group versus individual intervention moderated effects on the outcome - can you please explain the moderating effect?

We have now clarified that interventions were identified as being more effective if caregivers attended alone (i.e. without the person they cared for) and if the interventions were delivered in a group setting with other carers.

- Randomization was not included in the study design due to the input of the Patient and Public Involvement group - can you explain their concern with using randomization and how this decision was reached?

We have now provided this explanation in the PPI section.

- Part of the rationale for this study was the need for a longer-term follow up to examine long term benefits of the intervention. Please justify the choice of a 3-month follow up period and why this is considered long-term.

We stated at the end of the introduction that little research had focused on “longer” periods of time. We have not labelled our study as “long-term”. We have now provided a rationale for our selection of 12 weeks with 12 week follow-up which was based on pilot data from Wales (see Methods ‘participants’).

- With respect to the anxiety measure, what is the justification for the specified categorization as mild, moderate or severe?

These categorisations are not made by us but by validation studies of the scale. We have clarified this in the measures section and included a reference.

- The statistical analyses are well described and appropriate to address the research objective. Can you provide a sample size estimate for the univariable, multivariable, and sub-group analyses?

Unadjusted and adjusted analyses contained the same sample size and the numbers in sub-group analyses are all detailed in the results section.

- In table 1, considering the small sample size, please consider collapsing some of the categorical responses (e.g., income, employment status, singing confidence, length of time as carer, etc) and re-examine group equivalency. Also, clarify how these variables were included in the multivariable models. Were they included as is or in collapsed groups?

Following a suggestion from reviewer 3, we have now changed our statistical tests to Fisher’s exact rather than chi-square tests, which enable us to work with small sample sizes in different categories. These variables are included in their table 1 form in the multivariable models, which we feel is important in order to provide more sensitive measures of covariates.

- Clarify if the presentation of the results are based on the univariable or multivariable analyses. It would be beneficial to present these results in table form.

We understand that for the journal we are unable to duplicate information from figures in tables too, which is why we have provided statistical details in the text and graphs showing the changes over time. However, we have now clarified that results presented are all from fully-adjusted models.

- Please explain why the sensitivity analyses were conducted and why these specific analyses.

We have now clarified the rationale for conducting these chosen sensitivity analyses in the statistics section. Our first sensitivity analysis was focused on participants with poorer mental health at baseline in order to identify if those who had the most need of support benefitted or just those with good mental health already. Our second sensitivity analysis adjusted for singing patterns after 12 weeks as individuals made the choice of whether to continue or not and we wanted to model the effect of this choice on 24-week outcomes.

- With respect to the CONSORT diagram (Figure 1), please expand to provide more detail about the 330 potential participants who did not meet inclusion / exclusion criteria. What criteria did they not meet? Also include reasons participants were lost to follow up.

We have now updated Figure 1 to list the reasons for non-inclusion although we are unable to provide the precise numbers for these as some of our recruiting NHS sites did not provide the reason for screening ineligibility but just reported ‘ineligible’. However, we have clarified that ‘lost to follow-up’

meant that we could not make further contact with the participant and they stopped attending sessions and completing research measures.

- Figure 2, please clarify if this is a presentation of the unadjusted or adjusted analyses. It is not clear how significant group differences were identified when there are overlapping confidence intervals.

We have now clarified that the results shown in Figure 2 are adjusted for demographic factors, health-related factors, cultural engagement and attitudes to singing, and length of time caring for somebody with cancer. The significant differences were time by group interactions so confidence intervals can still overlap at specific time point, but the rate of change between time points is significantly different between groups.

Reviewer: 2

Reviewer Name: Dr. Zoe Hoare

Institution and Country: NWOORTH CTU - Bangor University - UK Please state any competing interests or state 'None declared': None declared

- Please leave your comments for the authors below The concept of using singing therapies is not new and has been evaluated and utilised in many participant groups. Unfortunately I think the design of this study has not contributed to the addition of new knowledge to the literature. There was no justification of why carers for cancer patients in particular rather than carers more generally, as this type of intervention has been used in dementia 'Singing for the Brain', 'Singing together'. Was this just a convenience sample?

We are grateful to Dr Hoare for her comments on this manuscript and suggestions, which are very useful and all of which we have addressed. In relation to the focus on cancer carers, our entire study rationale was to explore whether singing could help address known mental health challenges facing cancer carers. So this was not a convenience sample of cancer carers rather than other carers but rather cancer carers were our a priori focus. This rationale is outlined in the introduction where we make it clear right from the opening sentence that we were interested in cancer carers. However, we agree that there is other literature on carers and singing that needs referencing. We have now added sentences into the final paragraph of the introduction referencing some of the literature on singing and other carer groups such as dementia carers to show that this work on singing and cancer carers is not the only work on singing and carers generally. We have also noted in the limitations section that our results cannot at present be generalised to other carer groups.

- For me the glaring limitation of the design is in relation to not using randomisation, there may have been justified reasons for this but these are not clearly described in the paper alluding that PPI input agreed that the design should not be randomised is not enough. There are a number of designs that could have been considered that would have allowed consideration of participant preference. Allowing the participants to 'choose' their group will have significantly biased the estimate of effect as those who chose the intervention would have been motivated differently to those who did not (although it was interesting to see that the numbers who had had previous choral experience were higher in the control arm - possibly indicating that this type of intervention really wasn't for them). In my opinion, I don't think this bias is analogous with 'mimicking real life choice' as the effect estimate. In a pragmatic randomised design the effect estimate would be representative of the effectiveness of the intervention when offered to the specified population, however this preference design is self-selecting and potentially drastically increases the effect estimate produced. Those who choose not to

partake in the intervention are more 'extreme' to the possible advantage of the intervention than those possibly who would have signed up for a randomised study and been assigned an arm not particularly of their preference, i.e. allowing the preference has included participants that would never have signed up for this type of evaluation if it were randomised. Comparison of baseline factors to establish comparability does not take into account this preference/motivation difference.

We agree with this limitation of the study. As this was the first study of the cancer choirs at the Royal Marsden Hospital sites (or indeed in England) and there was therefore not data about the response from individuals in the community or the hospitals to the choirs, it was proposed by our PPI group that the study could be run as a longitudinal controlled study rather than a randomised controlled trial (RCT). As such, any carers would have the option of joining immediately rather than becoming part of a wait-list control. Whilst this has obvious methodological limitations, it was felt to be an important precursor to an RCT that would enable an assessment of the reception of the choirs in these new locations alongside measurements of outcome data. As such, we do not see this study as the 'end goal' but rather as a stepping stone towards larger and more robust future studies. In order to make this clearer, we have now stated explicitly that this study was non-randomised on the abstract and in the strengths and limitations bullet points so readers see immediately this limitation. Further, we have expanded on the PPI section to explain the rationale more clearly to readers. Although the study does have this design limitation, we would argue that the study has achieved its aim of providing evidence to suggest the feasibility of implementing the intervention and to suggest there can be benefits of the intervention for mental health in carers that can now be built on in a future randomised study. This study is an important stepping stone towards that.

- The sample size used appears to be convenience rather than justified and although acknowledged in the limitations explanation of the use of the sample would be useful in understanding the context of the design.

This was a convenience sample size as we did not have suitable data to perform a sample size calculation prior to recruitment based on a pre-calculated effect size. Our choice of 30 participants was based on predicted feasible recruitment numbers within the time available for the study and capacity within the choirs. As in our response above, this study was designed as a precursor to a larger study, so with the data we collected we could now assess the adequate sample size for a future study.

- Given the design it is reassuring to see that the models used to elicit the results include the known confounders (although given the model building technique this was not by plan it seems) and I agree that the statistical analysis used is appropriate for the data.

We are pleased that the reviewer agrees with the statistical analysis used.

- Whilst some statistical significance is found there appears to be no relation of this statistical finding to clinical impact? Given the small sample it is likely that this impact will be of clinical significance but in reporting this needs to be drawn out, particularly as this estimate is likely to be biased given the design.

Identifying clinically-meaningful changes from the validated measures we used can be tricky. However, we have now added in descriptive statistics showing the percentage change in anxiety and wellbeing across the 24 weeks so that readers can assess the magnitude of the improvement.

- The sensitivity analysis given in the paper would be better framed as sub-group analysis than sensitivity analysis as this is essentially what it is and given the sample size is even more limited in the conclusions that can be drawn.

We have now clarified that the analysis of those with poor mental health was a sub-group analysis. For the analysis that adjusted for singing behaviours after 12 weeks we have retained the label of sensitivity analysis.

- Although dropout from the study is considered and missing data accounted for via the statistical model one thing that has not been described or accounted for is attendance /adherence to the intervention. The assumption currently is that all participants who provide data until the 24 week follow up attend all sessions of the provided intervention.

We have now added in further detail on the adherence in the methods section, showing that participants attended an average of 10 out of 12 sessions and 57% of them had complete attendance.

- I don't agree that a strength of the study should be highlighted as using validated measures as it is unlikely that another type of measure would be used.

We have now removed this as a strength.

- I appreciate that the design is as given now and very little can be done about this and there are a number of justified reasons as to why this design would be appropriate. I do accept that a rigorously conducted longitudinal study can provide good evidence (even more than a poorly conducted randomised design in some circumstances) but the justification and reasoning needs to be clear in the reporting. If the design choices relating to the design and sample size can be justified then this will greatly help the explanation of the study findings, with appropriate addressing of the limitations.

We have explained the rationale of conducting the study as a non-randomised study in much greater our PPI section. The decision was that this study would be a precursor to a larger RCT that would enable an assessment of the reception of the choirs in these locations alongside measurements of outcome data. As such we hope that this paper can now be accepted for publication with these goals more clearly defined.

Reviewer: 3

Reviewer Name: Nick Garrett PhD

Institution and Country: Auckland University of Technology, Auckland, New Zealand Please state any competing interests or state 'None declared': None declared

- Please leave your comments for the authors below The article is well written (one typo one page 8 line 176 'different' should be 'difference') however the biggest design issue is the lack of randomisation between the intervention and control groups. This in itself while not ideal is acceptable and as the authors state mimicks real life, but needs to be more explicit stated in the abstract (just stated 'selected to join ...' is understating the lack of randomisation) and interpretation of results not just in the limitations at the end and especially as it is not stated in the abstract limitations.

We are pleased that Dr Garrett thinks the article is well-written. We realise that our lack of randomisation is a limitation and we have now clarified the design in the abstract, the strengths and limitations bullet point, in the methods and in discussion sections. We have also corrected the typo.

- It is not clear in the abstract or in the measures section that the HADS has both anxiety and depression subscales, it is only by either prior knowledge or inference by the fact that anxiety and depression are reported separately.

We have now clarified this in the measures section.

- There is a large reduction from those screened to the number that were eligible. It would be useful to readers to know the impact of each of the eligibility criteria on numbers i.e. whether it was therapy, medication, English language, or already singing.

Unfortunately we do not have the precise breakdown of screening reasons available as some NHS trusts who recruited to this study did not provide this full information on their screening but merely reported participants as 'eligible' or 'ineligible'. However, we have clarified the exclusion criteria on Figure 1 to make it clearer why the overall number of excluded participants was considered ineligible.

- A number of participants (42%) continued on with the choir, while it appears to have no impact based on the results, it would be ideal to know whether all of them continued past the 3 month follow-up point or not. If they did not all continue past the 3 month assessment then this would reduce the impact of no significant effect at 3 months. I also note that no numbers or statistics are reported for the difference for those that continued vs those that did not, with the reduction in numbers at this point the size of the effect is most probably more important than its significance.

We have now compared differences between those who did and did not continue singing and there are no significant differences in demographics or mental health (or any average differences that look of any importance). We have clarified this in the results section.

- The statistics section of the methodology states that "We modelled time both as a continuous variable and as a four separate time points". Is this in the same model, or separate models, it is not clear. Also the results specify time effects and group x time interactions, which time variable are these?

We have now clarified that time was modelled first as both a continuous variable to identify an overall time by group relationship (using a linear model as we found no evidence to support the use of a quadratic model), and then time was modelled as four separate time points to identify where specific changes occurred across these four time points. The results are reported in the same way (continuous results first and then the results at the four specific time points), but we hope this is now much clearer.

- The table of the baseline demographics of participants show a number of zero cells for the control group i.e. income >91,000, voluntary work and 0-3 months being a carer. These zero cells would have an impact on the appropriateness of a Chi-square test. Either a collapsing of categories or use of a Fisher's Exact test would be appropriate, and may have an impact on the significance of the test. It is interesting to note that the most wealthy, volunteers, and recent carers all selected to be in the intervention. These factors along with the confidence in signing, all show something slightly different about the intervention group.

Thank you for this suggestion. We have now used Fisher's Exact test for non-linear tests.

- The use of confounders in the model building was confusing, the methodology refers to the building up of the final model with demographics, health related, cultural engagement and attitudes to singing, and length of time caring. However nowhere is there reference to exactly which factors were actually included in the final models. Is it all the baseline factors regardless of whether they had statistically significant results or only a select few. Or were other factors not in Table 1 utilised this is not clear.

All the factors in Table 1 are in the analyses as covariates regardless of whether they were significantly related as they were determined to be potentially confounding factors that needed to be taken into account. We have now explicitly listed the covariates in the models in the penultimate paragraph of the statistics section where we outline the 5 models we built.

- All of the linear mixed effects models are presented in text only, and it is difficult to make sense of the different components of the model presented in the text. There appears to be an overall time x group interaction effect. Then some time point by time point effects/statistics are examined, however this is not referred to in the methodology. Similar format is used in the sensitivity analysis.

This links to point three above. We have now clarified throughout the results section when we were using time as a continuous variable (so looking at the overall time*group interaction) and when we modelled time as four separate time points (in order to identify at which specific points there were differences between the groups).

- For the strengths and limitations in the article summary, the first two dot points are design not strengths or limitations of the study.

We have clarified these two bullets to show how the design features were strengths of the study due to the methodological advancement on previous studies they provided.

Reviewer: 4

Reviewer Name: Stelios Zimeras

Institution and Country: University of the Aegean - Department of Statistics and Actuarial-Financial Mathematics Please state any competing interests or state 'None declared': None declared

Please leave your comments for the authors below The paper is interesting. Some statistical suggestions must firstly considered

1. The authors have not include analysis of normality. They consider probably normal distributions for their data that it is not correct. The have applied parametric statistical analysis but they have not investigate if their data are ok for that.
2. The number of patient are very low. 29 for control and 33 for choir. They have not investigate the sampling significance of their data. They are ok for their analysis.
3. Based on their data the reader would wait an logistic regression analysis or OR analysis

We can confirm that we tested all model assumptions including running tests of normality of standardised residuals prior to conducting our analyses. We have clarified this in the statistics section. We acknowledge that the participant numbers are low but are glad that there is agreement that they are OK for the analysis. We agree that we could now conduct further analyses including logistic regression analyses but we feel there is already a lot of information within the paper so propose leaving any further analyses for now.

VERSION 2 – REVIEW

REVIEWER	Jill Cameron University of Toronto, Canada
REVIEW RETURNED	23-Apr-2019

GENERAL COMMENTS	This is an interesting paper and reads well. The authors have done a nice job addressing reviewer comments. Please consider the following points: The analyses are quite specific suggesting that hypotheses were tested. Can the authors include a more specific objective in the abstract and include the hypotheses tested in the body of the manuscript? Were the sub-group analyses determined a priori? I think there is a typo in the results compared to the discussion where the sub-group analysis related to well-being suggests "above average" versus "below average" in the discussion. Clarify. Please provide the sample size estimates to demonstrate that the sample size is sufficient to support the overall and sub-group analyses. How was missing data addressed in the sub-group analyses? In the results, section, do the baseline rates of anxiety differ between the intervention and control groups? In the article summary and conclusions, include that future studies should involve a RCT design. In the CONSORT diagram, include the number screened and the number eligible - they seem to be missing from the diagram. Can you also include reasons participants were lost to follow-up from baseline through week 24? Overall, findings need to be presented more cautiously as this was a non-randomized design with a small sample size. A larger scale RCT is needed to demonstrate benefits.
---

REVIEWER	Dr. Zoe Hoare NORTH CTU, Bangor University, UK
REVIEW RETURNED	12-Apr-2019

GENERAL COMMENTS	I thank the authors for their through responses to my and other reviewers comments, given the adaptations to the article I think the contextual setting is much more apparent for the resarch which makes it much more justifiable. Couple of very minor points Under participants and procedure bottom of page 5 and further down on page 6 there are some results included (CONSORT description and attendance post 12 week) - these should be moved to the results section rather than in the methods section. Table 1 includes a footnote C bt with no actual associated footnote.
---

	Top end of the CONSORT flowchart implies that no-one eligible declined to take part (386 screened of which 66 were eligible....) - is this true and if so do the authors think the design adopted influenced this?
--	--

REVIEWER	Nick Garrett PhD Auckland University of Technology New Zealand
REVIEW RETURNED	23-Apr-2019

GENERAL COMMENTS	The present draft is acceptable (not exceptional) but there are two things - Under measures the anxiety and depression measures are subscales of the HADS not scales - The key baseline finding of significant differences in confidence in singing is a major consideration for this study i.e. those that do not select the singing intervention do not have confidence in singing. This is a major factor in the self selection process but has not been addressed in the discussion section (namely that the intervention may not be appropriate for all caregivers).
---

REVIEWER	Stelios Zimeras University of the Aegean, Dept. of Statistics and Actuarial-Financial Mathematics
REVIEW RETURNED	08-Apr-2019

GENERAL COMMENTS	Fine analysis
---------------

VERSION 2 – AUTHOR RESPONSE

Response to Reviewers

Reviewer: 4 | Reviewer Name: Stelios Zimeras | Institution and Country: University of the Aegean, Dept. of Statistics and Actuarial-Financial Mathematics

Please state any competing interests or state 'None declared': None declared

Reviewer: 2 | Reviewer Name: Dr. Zoe Hoare | Institution and Country: NORTHWORTH CTU, Bangor University, UK

Please state any competing interests or state 'None declared': None declared

I thank the authors for their thorough responses to my and other reviewers comments, given the adaptations to the article I think the contextual setting is much more apparent for the research which makes it much more justifiable. Couple of very minor points

- Under participants and procedure bottom of page 5 and further down on page 6 there are some results included (CONSORT description and attendance post 12 week) - these should be moved to the results section rather than in the methods section.
- Table 1 includes a footnote C bt with no actual associated footnote.
- Top end of the CONSORT flowchart implies that no-one eligible declined to take part (386 screened of which 66 were eligible....) - is this true and if so do the authors think the design adopted influenced this?

***We're grateful to Reviewer 2 again for her comments on this paper. We actually disagree that the figures of attendance are results – we argue those show our participant selection for sample size so more suitable for the methods section. However, we have corrected the footnote 'c' error. Regarding the eligible, our CONSORT diagram and our text in the methods section shows that 4 participants who were eligible declined to take part or were lost to follow-up prior to the study commencing. We have now altered the text in Figure 1 to clarify this and also in the text.

Reviewer: 3 | Reviewer Name: Nick Garrett PhD | Institution and Country: Auckland University of Technology New Zealand

Please state any competing interests or state 'None declared': None declared

The present draft is acceptable (not exceptional) but there are two things

- Under measures the anxiety and depression measures are subscales of the HADS not scales
- The key baseline finding of significant differences in confidence in singing is a major consideration for this study i.e. those that do not select the singing intervention do not have confidence in singing. This is a major factor in the self selection process but has not been addressed in the discussion section (namely that the intervention may not be appropriate for all caregivers).

***We'd like to thank Reviewer 3 for his comments. We have updated the HADS description to say subscales and also included the additional limitation as requested.

Reviewer: 1 | Reviewer Name: Jill Cameron | Institution and Country: University of Toronto, Canada

Please state any competing interests or state 'None declared': none declared

This is an interesting paper and reads well. The authors have done a nice job addressing reviewer comments. Please consider the following point. The analyses are quite specific suggesting that hypotheses were tested. Can the authors include a more specific objective in the abstract and include the hypotheses tested in the body of the manuscript? Were the sub-group analyses determined a priori? I think there is a typo in the results compared to the discussion where the sub-group analysis related to well-being suggests "above average" versus "below average" in the discussion. Clarify.

***It is correct we had specific hypotheses and we have now clarified these in the text introduction and discussion. The subgroup analyses were determined a priori. In the results, we explain that we excluded people with above average wellbeing, whereas we then say in the results and discussion that we found strongest responses for those with below-average wellbeing, so this is not a typo.

Please provide the sample size estimates to demonstrate that the sample size is sufficient to support the overall and sub-group analyses. How was missing data addressed in the sub-group analyses?

***As discussed in our previous response to the other reviewers, we did not have suitable data to perform a sample size calculation prior to recruitment based on a pre-calculated effect size. Our choice of 30 participants was based on predicted feasible recruitment numbers within the time available for the study and capacity within the choirs. This study was designed as a precursor to a larger study, so with the data we collected we could now assess the adequate sample size for a future study. We used LMMs as our statistical approach. Unlike some other repeated measures analysis models such as repeated measures analyses of variance which deal with missing data through list-wise deletion, linear mixed effects models make full use of the dataset, so participants who dropped out were still included in the subgroup analyses using the available data we had on them. This is outlined on page 7 in the statistics section.

In the results, section, do the baseline rates of anxiety differ between the intervention and control groups?

***The baseline results did not differ for any of the mental health measures at baseline. We have clarified this in the text as well as in the table.

In the article summary and conclusions, include that future studies should involve a RCT design.

***We have now added this as requested.

In the CONSORT diagram, include the number screened and the number eligible - they seem to be missing from the diagram. Can you also include reasons participants were lost to follow-up from baseline through week 24?

***Thank you for noticing the missing numbers. When we converted to JPEG they were lost due to the colour of font. We have now added them back in. Lost to follow-up meant that we could not make further contact with the participant and they stopped attending sessions and completing research measures. So we do not know why this was as we could not follow them up.

Overall, findings need to be presented more cautiously as this was a non-randomized design with a small sample size. A larger scale RCT is needed to demonstrate benefits.

***We have now gone through the manuscript nuancing our findings, in particular clarifying that we cannot confirm causality given the lack of randomisation. We have also clarified at the end of the discussion that the next step is an RCT to take this further.